# International Perspectives of Extended Genetic Sequencing When Used as Part of Newborn Screening to Identify Cystic Fibrosis

**DOI:** 10.3390/ijns10020031

**Published:** 2024-04-08

**Authors:** Corinna C. A. Clark, Pru Holder, Felicity K. Boardman, Louise Moody, Jacqui Cowlard, Lorna Allen, Claire Walter, James R. Bonham, Jane Chudleigh

**Affiliations:** 1Warwick Medical School, Warwick University, Coventry CV4 7AL, UK; felicity.boardman@warwick.ac.uk; 2Florence Nightingale Faculty of Nursing, Midwifery & Palliative Care, King’s College London, London SE5 9PJ, UK; pru.holder@kcl.ac.uk (P.H.); jane.2.chudleigh@kcl.ac.uk (J.C.); 3Centre for Arts, Memory and Communities, Coventry University, Coventry CV1 5FB, UK; l.moody@coventry.ac.uk; 4Paediatric Respiratory Medicine, Royal London Children’s Hospital, London E1 1FR, UK; 5Cystic Fibrosis Trust, London EC3N 1RE, UK; 6Pharmacy, Diagnostics and Genetics, Sheffield Children’s NHS Foundation Trust, Sheffield S10 2TH, UK; j.bonham@nhs.net

**Keywords:** cystic fibrosis, next-generation sequencing, genomics, CRMS/CFSPID

## Abstract

There is increasing interest in using extended genetic sequencing (EGS) in newborn screening (NBS) for cystic fibrosis (CF). How this is implemented will change the number of children being given an uncertain outcome of CRMS/CFSPID (cystic fibrosis transmembrane conductance regulator (CFTR)-related metabolic syndrome/CF Screen Positive Inconclusive Diagnosis), probable carrier results, and the number of missed CF diagnoses. An international survey of CF health professionals was used to gather views on two approaches to EGS—specific (may reduce detection of CRMS/CFSID but miss some CF cases) versus sensitive (may increase detection of CRMS/CFSPID but avoid missing more CF cases). Health professionals acknowledged the anxiety caused to parents (and health professionals) from the uncertainty surrounding the prognosis and management of CRMS/CFSPID. However, most preferred the sensitive approach, as overall, identifying more cases of CRMS/CFSPID was viewed as less physically and psychologically damaging than a missed case of CF. The importance of early diagnosis and treatment for CF to ensure better health outcomes and reducing diagnostic odysseys for parents were highlighted. A potential benefit to identifying more children with CRMS/CFSPID included increasing knowledge to obtain a better understanding of how these children should best be managed in the future.

## 1. Introduction

Advances in genomic sequencing technologies mean that the use of extended genetic sequencing (EGS) has become feasible for cystic fibrosis (CF) newborn screening (NBS). CF is caused by pathogenic variations of the CF transmembrane conductance regulator (CFTR) gene. Using EGS would greatly increase the number of CFTR variants included in the NBS algorithm and has the potential to reduce the false-positive rate and the number of repeat samples [1]. How EGS is applied, however, will have consequences for the number of newborns receiving an inconclusive result or, potentially, the number of missed diagnoses at screening, both important considerations for screening programmes.

Newborn screening algorithms for CF differ internationally [1,2]. First-tier testing generally consists of measuring immunoreactive trypsinogen (IRT). Second-tier testing can include IRT, pancreatitis-associated protein (PAP), and/or DNA analysis using differing compositions of variant panels [1]. A small number of screening programmes have implemented EGS, as a third tier [3]. Following a positive NBS result, the most reliable and widely available diagnostic test is the sweat chloride (SC) test [4]. While most children with a positive NBS result will be confirmed as being affected by (or carriers of) CF using the SC test, a number of infants will receive an inconclusive outcome—termed cystic fibrosis transmembrane conductance regulator (CFTR)-related metabolic syndrome or CF Screen Positive Inconclusive Diagnosis (CRMS/CFSPID) [5]. Children with CRMS/CFSPID have either a normal sweat chloride (<30 mmol/L) and two CFTR mutations (at least one of which has unclear phenotypic consequences) or an intermediate sweat chloride value (30–59 mmol/L) and one or no CFTR mutations [6,7]. The incidence of CRMS/CFSPID varies internationally depending on the population and algorithms used. It is unclear how many of these children will go on to develop CF or a CFTR-related disorder [8,9,10,11,12,13,14]: the latter is defined as *“…clinical conditions that are recognised to be associated with abnormality of the CFTR gene but are not CF”* [7]. Depending on how EGS is used (Box 1), it could result in more children with CF being missed (undiagnosed following NBS) but fewer children with CRMS/CFSPID being identified (a specific approach), or fewer children with CF being missed but more children with CRMS/CFSPID being identified (a sensitive approach) [3].

Box 1Explanation of sensitive versus specific approach used in the survey.**Referring babies with a combined score of 3** (one pathogenic variant combined with one variant of varying clinical consequence) wouldResult in an increase (from 25 per annum to 80 per annum in the UK) in the designation of infants with CRMS/CFSPID.Be likely to help avoid a small number of missed CF cases when compared with restricting reporting to those with a score of 4.
**Referring babies with a combined score of 4** (or two clearly pathogenic variants) wouldResult in a reduction (from 25 per annum to 5 per annum in the UK) in the designation of infants with CRMS/CFSPID.Create a possible chance (less than 10 per annum) of those with true CF being missed at screening—the majority of these babies will be diagnosed clinically by the age of two years.

A global harmonization process was undertaken in 2016, to provide a consistent international approach and definition for CRMS/CFSPID. This resulted in the development of international guidance for the management of this designation [7].The study presented here sought to explore international perspectives of health professionals on the use of EGS as part of CF NBS and explore health professionals’ experiences of caring for children with CRMS/CFSPID. A survey was designed to explore views and opinions on the proposed CF screening protocol incorporating EGS vis-à-vis current UK practice, including how these options impact identification of CRMS/CFSPID and missed cases of CF. We have used this method successfully previously to explore the management of CF [2] and CRMS/CFSPID [15]. This formed part of a larger package of work, which included the views of adults living with CF and families of children with CF and CRMS/CFSPID [16,17].

## 2. Materials and Methods

### 2.1. Development and Design of the Online Survey

An online survey was developed with input from five professionals with extensive experience of working in the CF field. These included a representative of the UK CF Trust, a CF nurse specialist, a children’s nurse, a CF medical consultant and the Laboratory Lead for the UK NBS Programme Centre.

The first part of the survey contained five demographic questions and asked for the country and region in which the professional worked, their role (16 options to select from or respondent could self-declare profession), professional qualifications and number of years’ experience working with children/adults/families with CF (see Appendix A for full survey). After providing a brief summary on the current screening protocol and the potential impact of using ESG (see Box 1), the survey asked for views on the sensitive and specific approaches utilising six closed questions and five open-ended questions. The closed questions used a 5-point Likert response scale from very important to not at all important or sliding scale questions (0 not at all important to 10 very important). The final section was only presented to professionals who stated they had experience of caring for children with CF and their families. This section included four closed questions and two open-ended questions on the impact of a CRMS/CFSPID designation on families, and also questions relating to their experience of current CRMS/CFSPID treatment pathways. Survey participants were also given the opportunity to provide free text comments in response to these questions.

### 2.2. Administration of the Survey

The survey was administered via Qualtrics (www.qualtrics.com) and was available between June 2022 and January 2023. Participants were self-selecting, with inclusion criteria being that they were involved in processing and/or communicating positive NBS results for CF to families, or supporting families in health settings. Recruitment was achieved through social media, national and international professional societies and targeting known CF professionals.

### 2.3. Data Analysis

Participant location was categorized into UK, United States (US) and Europe/Rest of World (RoW). Years working was categorized (5 years and under, 6 to 10 years, 11 to 20 years, 21 to 30 years, and over 30 years).

Quantitative survey data were summarized and analysed using IBM SPSS version 28.0.1.1. For questions that used a Likert scale (low/no importance, neutral, moderately important, very important) frequency of responding by region was compared using Chi Squared test; where the assumptions of Chi were validated, Fisher-Freeman-Halton Exact Test was used. Responses using a sliding scale from 1–10 were analysed using Kruskal–Wallis with pairwise comparisons between regions (using a Bonferroni correction).

Qualitative open-ended text responses were analysed thematically [18] into broad themes and sub-themes; the number of responses that referred to each of these themes was then quantified for each question.

## 3. Results

### 3.1. Responses

There were 189 submissions, of which 81 were excluded as participants had only answered the initial questions about their location and profession (of these 28 were UK based, 14 from Europe/RoW, 23 from the US and Canada, and 16 did not specify). A further 7 participants (4 UK based, 3 Europe/RoW) stopped responding by the fourth screening question; these responses were included in the analysis for the questions they responded to.

In total, there were 108 usable responses (47 UK, 27 Europe/RoW, 34 US) to the first two sections of the survey. Of the Europe/RoW category, 5 responses came from non-European countries (New Zealand, Turkey, Argentina, Brazil). Responses were not compulsory and therefore the denominator varies by question. A total of 84 participants answered yes to the question ‘do you have experience of caring for children with CF and their families’ and were asked the questions in the final section (impact of a CRMS/CFSPID designation/delayed CF diagnosis, and treatment pathways).

### 3.2. Demographics

Participants described their professional roles using 17 job titles (Table 1).

Professions were not evenly spread across the regions, for example, CF medical consultants were the largest single category overall (31% of all responses) and in responses from the UK (40%) and Europe/RoW (33%). However, CF medical consultants only made up (18%) of participants from the US, where the largest single category were Centre Directors (29%). CF Clinical nurse specialists constituted 28% of the responses from UK health professionals, but they did not feature in the data from Europe/RoW and constituted 9% of the participants from the US.

Overall, 61% of participants (62% in UK) had at least 15 years of experience working with children/adults/families affected by CF (including carriers and CRMS/CFSPID) and most were involved in the care of children with CF and their families (78% overall, 70% UK) (see Table 2).

A summary of responses to the second section of the survey are summarized in Table 3 and Table 4. Further details and responses to open-ended questions and comments are summarized below.

### 3.3. The Importance of Not Missing Babies with True CF

Most participants (86% overall) thought it was very important to ensure babies with CF are not missed during screening. The proportion of participants who thought that this was very important, rather than moderately important differed between regions (*p* = 0.004). Although most UK participants selected very important, a greater proportion selected moderately important (very important 77%; moderately important 21%) compared to participants from Europe/RoW (very important 85%; moderately important 7%) and the US (100% very important) (Table 3).

Comments by health professionals in response to this question most frequently referenced there being poorer clinical outcomes from a delayed diagnosis (54 mentions in 72 comments).
“*The earlier treatment is started in true CF the outcome for the individual will be better in terms of lung function, nutritional benefit, and well-being. A missed diagnosis delays treatment options and children are more likely to have pathogens in their airways and potential hospital admissions. It may delay the start of modulator therapy. The psychological impact should also be considered of a late diagnosis picked up if a child is symptomatic rather than through NBS.*”(CF Clinical Nurse Specialist, UK)
“*Profound impact on early growth if missed. Early recognition saves multiple visits to the physician for growth measurement and support, as well as significant parental stress during what is already an exhausting time. Potential impact in the future should HEMs become available at younger ages…*”(CF Consultant, US)

The psychological impact on families of a delayed or missed diagnosis was also highlighted by several participants (7/72). Other themes frequently mentioned included the importance of maximizing diagnostic rate of CF cases (12/72) and analogous to this, reference to the purpose of newborn screening being to detect babies with CF (11/72).
“*A primary objective of NBS is to optimize sensitivity for early diagnosis.*”(Genetic counsellor, US)
“*Newborn screening has definitively been proven to result in better outcomes. Choosing an algorithm that is known to miss infants with CF makes no sense.*”(Centre Director, US)

Several participants commented on the risk of increasing the diagnostic odyssey and potential for poorer health outcomes where CF is missed at newborn screening, as clinical teams will assume that a negative screening result for CF would be definitive (9/72).
“*Whilst experience of clinically diagnosing CF was the norm prior to the introduction of NBS, this has waned in the intervening years. Clinicians have likely dropped their guard when it comes to requesting sweat testing and I worry that the changes to NBS will be too subtle to be understood by all parties…*”(CF Consultant, UK)
“*Missing babies with false negative newborn screening may lead to delayed diagnosis, because pediatricians are not that aware of CF after the negative screening, diagnosis by clinical symptoms always means problems and organ damage you might not be able to resolve again*”(CF Consultant, Europe/RoW)
“*Critical to initiate treatment, particularly enzymes to avoid failure to thrive and improve long-term outcomes. Also, hopefully more babies eligible for modulators before long, which could have a significant impact. Kids with symptoms but normal NBS are more likely to face a ‘diagnostic odyssey’ and have delay in care if providers and parents think CF is unlikely. Detecting those affected is the core purpose of NBS.*”(Genetic Counsellor, US)

### 3.4. The Importance of Reducing the Number of Babies Reported with a CRMS/CFSPID Designation by Using a Specific Approach, Compared with Increasing the Number Reported Using a Sensitive Approach

In the context of reducing the number of babies with a CRMS/CFSPID designation, 71% of participants thought that this was very or moderately important (62% UK, 92% Europe/RoW, and 67% US). Proportionally more participants from Europe/RoW (65%) thought that this was very important, compared with the UK (40%) and US (40%), but this did not reach significance (Table 3).

Half of all comments in response to this question acknowledged the psychological burden on families from a CRMS/CFSPID designation (mentioned 31 times in 62 comments) and there was concern around the medicalization of children with a CRMS/CFSPID designation (13/62).
“*The CFSPID categorization causes a lot of anxiety and need for follow-up with no obvious benefit.*”(CF Consultant, UK)
“*Oftentimes babies with CRMS are generally healthy and followed due to abnormal genetic results or sweat testing. The guidelines on care for these children is unclear. Many of these are likely overmedicalized- followed up in CF clinic with throat cultures or other testing for some unknown number of years … when they likely will be healthy and would not have presented for medical care (to a CF center) in childhood otherwise. This can be a significant burden on families and medical facilities*”(Centre Director, US)

Eleven health professionals remarked on the confusing nature of a CRMS/CFSPID designation:
“*I think it’s hard for parents to get a diagnosis for CFSPID as they don’t really understand what it means. It’s hard enough for professionals to completely understand it too.*”(CF Clinical Nurse Specialist, UK)
“*CFSPID Diagnoses are very stressful to families and providers alike. It will take ALOT of education to CF teams on how to manage these patients- and keep track of them.*”(CF Clinical Nurse Specialist, US)

Many participants who mentioned the psychological burden of an uncertain diagnosis, also commented on the benefits of monitoring these children should they later develop CF or a CFTR related disorder (16/62). A small number commented on the accrued knowledge over time from detecting CRMS/CFSPID cases (3/62) and several indicated that, on balance, not missing true CF should take priority over reducing CRMS/CFSPID cases (10/62).
“*These babies are challenging, and time consuming to manage and emotionally challenging for families to cope with so it would be “nicer” to be presented with fewer of them HOWEVER, the more we come across the more knowledge we will accumulate and they will become less challenging over time. I think important to tackle head on and gather as much info as possible to inform best practice for the future.*”(Pediatrician, UK)
“*At the moment we don’t know enough about the long term outcome of CFSPID, so we can´t say w[h]ich baby will turn to the diagnose CF [sic] later on—with good communication it might be better, to see also those babies.*”(CF Consultant, Europe/RoW)

Several health professionals suggested that improving the management of these cases was key to reducing anxiety in families and over-medicalization (10/62).
“*Whilst a CFSPID diagnosis can be difficult to accept for families, the priority here should be to improve in the management of these children in order to avoid the over-medicalization that has gone before. I also believe that, as time goes on, those mutations considered to be of uncertain significance will become better categorized.*”(CF Consultant, UK)

### 3.5. Importance of Using a Sensitive Approach

Overall, 80% thought it was important to use a sensitive approach, meaning greater reporting of CRMS/CFSPID. This was despite 71% of participants considering it important to limit reporting of CRMS/CFSPID by using a sensitive approach. There was no significant difference by region.

The most frequently discussed theme in response to this question focused on the importance of monitoring children with CRMS/CFSPID because of the likelihood that some would develop CF or a related disorder (19/42), followed by the need to prioritise diagnosis of true CF over reducing cases of CRMS/CFSPID (9/42).
“*As an adult physician we see people presenting later in life with CF related disorders which may have been acted on sooner if they were identified as being at risk at birth.*”(CF Consultant, UK)
“*…we don’t yet know the lifetime risk for things like adult onset bronchiectasis, male infertility,* etc. *The result also provides important genetic information for family members who may be carriers or affected to varying degrees. I do think effort should be dedicated to how to inform, educate and follow these kids without over-stressing the family. For the kids who do convert to CF diagnosis, hopefully they’ll know the team and be empowered to start appropriate treatment early in the disease course.*”(Genetic Counsellor, US)

### 3.6. Importance of Using a Specific Rather Than Sensitive Approach

Fifty one percent of participants supported a specific approach. This was fewer than the 71% of participants who had supported this option when it had been framed in the context of reducing the number of children receiving a CRMS/CFSPID designation. Comparing responses to these two questions, 47% of participants selected that it is important to use a specific approach in both.

The most frequent themes referenced in the open-ended responses to this question were the importance of not missing cases of true CF and prioritizing this over reducing CRMS/CFSPID identification (14/40), as well as the importance of monitoring children with CRMS/CFSPID for changes in their health over time (9/40).
“*Reducing ambiguous outcomes should be a secondary goal to maximizing diagnostic rate.*”(Genetic Counsellor, US)
“*I do not find sharing the information of potential disease to be harmful. In fact, holding back the information may be detrimental to overall individual health, as they may take longer to put subtle clues to their diagnosis together.*”(CF Consultant, US)

### 3.7. The Importance of Reducing or Avoiding the Number of Babies Being Reported as ‘Probable CF Carriers’

Overall, 62% of health professionals thought that it was either moderately important or very important to reduce or avoid reporting probable carriers. There was an effect of region, with a greater proportion of participants from Europe/RoW (67%) seeing the avoidance of reporting probable carriers as very important, compared with those from the UK (21%) and US (18%).

The most frequent theme in the open-ended responses was the view that carrier information was seen by families as desirable and something many would choose to have (13/43). It was described as useful for reproductive knowledge and decision-making on 11/43 occasions.
“*…families like to know this information when it is shared—impact the child future if planning pregnancy of their own… families plan for future children with cascade testing*”(CF Clinical Nurse Specialist, UK)

The potential for psychological harm was acknowledged by some (11/43) as well as the risk of medicalization and not being the purpose of NBS (4/43) and the ethical complexities around the rights of the child not to know this information (4/43).
“*Stressful for families, difficult for GPs to manage, results in a demand to see respiratory pediatricians. Ethically controversial—these babies did not consent to testing and may not wish to have this information as they grow up.*”(Pediatrician, UK)
“*Unless there is a plan for extensive family cascade screening/testing most people would consider this a harm (genetic test results on a non-consented minor with no immediate health benefits to the tested child).*”(Newborn Screening Coordinator, Europe/RoW)

While some commented that carrier status has no clinical benefit (5/43), others suggested that detecting carrier status was beneficial as it is not necessarily a benign state (6/43):
“*I think we do not know what the effect of so-called carrier status is, and future research may identify an effect on CFTR function so just as long as this is explained to parents knowing they are carriers is information that is important*”(CF Consultant, UK)
“*Being a CF carrier has been shown not to be entirely benign. Genetic counseling for carriers and their families is useful. Ultimately this may be minimally harmful or potentially helpful.*”(Pediatrician, US)

### 3.8. Participant Ratings (Sliding Scale) of the Importance of Factors Implicated in the Use of EGS in CF NBS

Avoiding missing a baby with CF was seen as very important across all regions, with the median score of importance being the maximum of 10 (Table 4). Participants from the UK used a greater range of responses (5–10) compared with those from the US (9–10).

Avoiding identifying babies with CRMS/CFSPID was seen as quite important (median score UK and US 7, Europe/RoW 8), followed by avoiding reporting probable carrier status (UK 6, Europe/RoW 8, US 6), and reducing repeat IRT tests (UK 5, Europe/RoW 7, US 7) (Table 4).

There was some inconsistency in responding: of the 46 participants who rated the importance of avoiding identifying CRMS/CFSPID as a score 8 or above, 32 (70%) had thought it was important to ‘use a sensitive approach’. This may be attributable to either confusion in understanding the parameters of the choices they were being asked to make, or conflict between the desire to reduce CRMS/CFSPID designations and not miss identifying any true CF cases.

### 3.9. Responses to Questions concerning the Impact of CRMS/CFSPID, among Health Professionals with Experience of Caring for Children with CF and Their Families

Around 80% of participants thought that a CRMS/CFSPID designation impacted on families, with all but one of the remaining participants selecting that it ‘sometimes’ impacts on families (Table 5).

Most comments focused on the psychological burden to families of a CRMS/CFSPID designation (55/68). Some participants also commented on the medicalization of these children (10/68).
“*Very unsettling and often difficult to understand. Increased worry around the usual coughs and colds children get both from family and the CF/resp team*”(CF Consultant, UK)
“*Increased medical management for child, possible anxiety/confusion about uncertainty of dx[diagnosis], which can be detrimental to bonding with infant*”(Genetic Counsellor, US)

The importance of being aware of these children so that they could be monitored was also frequently discussed (15/68) as was the sense that families became better able to deal with this information over time (9/68).
“*Difficult [for] families to understand initially but this changes with time. There is more difficulty with older children where they have to be undiagnosed. There is worry and concern, but time allows the medical profession to learn how to manage these difficulties, rather than change the criteria so that we can potentially ignore this population group*”(CF Consultant, UK)
“*They know to watch out for clinical changes but most get on with life*”(CF Clinical Nurse Specialist, UK)

### 3.10. Frequently of Review for Children with a Designation of CRMS/CFSPID and Their Families

Sixty seven percent reported that children were either reviewed annually or 6-monthly. In the US and Europe/RoW, nearly half reported that children were reviewed annually (47% Europe/RoW, 45% US), compared with 29% in the UK, whereas a similar proportion across regions reported 6-monthly reviews (UK, 29%, Europe/RoW 29%, US 26%) (Table 5).

Of the 23 participants who detailed a different review frequency, 21 could be categorized as either depends on clinical signs/repeat sweat test (10%), frequently in first year then annually (8%), or variable between 2–6 times pa (9%). One respondent suggested that there was no ongoing review beyond 1–2 years (US), and another that ‘they are often lost to follow-up’ (US) (Table 5).

In the UK (58%) and US (55%), over half of participants reported that children were sometimes started on CF care pathways (Europe/RoW 41%), and this treatment did, or sometimes could, include CFTR modulators (UK 48%, US 38%) but this was rarely reported by participants from Europe/RoW (6%) (Table 5). Of the 42 participants who selected yes/sometimes in response to whether children are started on CF care pathways, 29 (69%) added a comment to clarify that this depended on the health of the child (including clinical symptoms and growth) and/or the results of clinical tests (e.g., cough swabs and lung health).

### 3.11. Impact of a Delayed CF Diagnosis

Participants focused on the potential for negative clinical outcomes for the child resulting from delayed diagnosis (65/68). Many participants referred to long-term and potentially irreversible damage to organs, the trauma from diagnostic delays, poor quality of life, and reduced life expectancy, as well as the potential to miss out on early treatment with modulator therapies.
“*In our experience all of our patients with delayed diagnosis have had higher morbidity and poorer outcomes.*”(CF Clinical Nurse Specialist, UK)
“*We know that children diagnosed earlier, who can begin on adequate treatments, will do better. In the new era of modulators, which will be life changing as children can start on these at younger and younger ages, it is of the utmost importance to diagnose these children early.*”(Centre Director, US)

A number of participants also commented on the psychological impact of a delayed diagnosis (16/68), as delayed diagnosis may be harder to accept, leading to issues such as poor engagement with clinical care and mistrust in the medical system (3/68).
“*Short term—could lead to poor growth and nutrition which in turn could impact development. Reduced lung function. Earlier acquisition of Pseudomonas aeruginosa which could have long term impact on respiratory health. All of these could impact long term outcome and life expectancy. From a practical point of view, it is also easier to establish a treatment regimen from an earlier age. If diagnosis is made later e.g., toddler years it can be very difficult to establish chest physio regimen, for example. If even later, e.g., teenage years—lots of challenges and often difficulty accepting diagnosis.*”(Pediatrician, UK)
“*One thing I’ve noticed is that the later the child is diagnosed, the longer and harder it is for parents to accept the diagnosis and then that leads to lower adherence to therapies which leads to suboptimal clinical outcomes.*”(Newborn Screening Coordinator, US)

Similarly, the predominant theme in response to the impact on the family was the psychological impact of a delayed diagnosis (50/64), again, with implications for a loss of trust in health professionals (15/64) and reduced adherence to treatment.
“*Could lead to feelings of anger, distrust in medical care, guilt. Difficulty accepting diagnosis and establishing treatment regimen at home.*”(Pediatrician, UK)
“*Having an undiagnosed ill child can be worse than treating a diagnosed child. A chronically ill undiagnosed child is a stigma of no small consequence for the parents.*”(Centre Director, US)

Also mentioned by some participants were the impacts on reproductive choices (10/64), either through the negative psychological impact of a delayed diagnosis affecting future reproductive choices (i.e., deciding not to have another child), or parents who had already embarked on a subsequent pregnancy without being aware of their ‘genetic risk’.

## 4. Discussion

This project used an online survey to explore international perspectives of health professionals on EGS when used as part of NBS to identify CF and the resultant impact on the number of children with a CRMS/CFSPID designation. Survey results were analysed by region (UK, Europe/RoW, US) to determine whether any differences in management approaches remain. Usable responses were obtained from 108 professionals, including 47 from the UK, 27 from Europe/RoW, and 34 from the US. Responses indicated few differences between regions.

The vast majority (97%) of health professionals thought it was important not to miss cases of true CF and 80% thought it was important to use a sensitive approach to EGS if it were to be introduced into the NBS for CF. This finding is analogous to the views of others with lived experience of CF [16,17], but is in contrast to previous work with the general public, who favoured a specific approach [19]. Other studies have found the general public have differently calibrated barometers of condition severity when compared to people with direct lived experience [20,21], which could explain this difference. It was evident that for many health professionals, the main reason for preferring the sensitive route was because of the reduced risk of missing CF cases compared with the specific route. Both the health implications and the emotional impact of delayed diagnoses were frequently mentioned, as well as experiences with reduced treatment compliance. Allied to these concerns was the worry that other health care professionals would not be aware that EGS could still miss cases, therefore dismissing CF as a potential diagnosis, leading to further diagnostic delays.

Health professionals acknowledged that due to the uncertainty associated with a CRMS/CFSPID designation, it can be an anxiety-provoking outcome for parents following newborn screening. This is supported in the literature, which indicates that detection of CRMS/CFSPID can be a worrisome result; studies suggest relief at not having a definite CF diagnosis are offset by confusion and uncertainty about the meaning of an inconclusive result with the potential for long-term harm [22,23]. Current guidance recommends that communication of CRMS/CFSPID results to parents should acknowledge the challenging situation CRMS/CFSPID presents [7] and consensus guidelines provide specific suggestions for genetic counselling for families following a CRMS/CFSPID designation [24]. Some health professionals in this study felt that anxiety associated with a CRMS/CFSPID designation may subside over time as the child continued to be well, and this ‘biological recalibration’ by parents based on their child’s good health has been reported elsewhere [25]. It is generally considered that most children with a CRMS/CFSPID designation will remain well. The reported proportion of children who convert from CRMS/CFSPID to a CF diagnosis varies from 2–48% [8,9,10,11,26,27,28,29,30,31], but in most studies this was below 10% [32].

While the definition of CRMS/CFSPID has been harmonized [6], it is acknowledged that it is still a complex designation [33]. In the current work, there was evidence of inconsistency in responding to the survey questions. This suggested either conflict in balancing the desire to reduce the ambiguity from CRMS/CFSPID designations with the desire not to miss any true cases of CF, or confusion around the questions relating to only using a specific approach (and not reporting CRMS/CFSPID), or a lack of understanding of the parameters of the choice they were being asked to make (despite this being described in the question stem, see Appendix A). It is acknowledged this may have influenced some of the responses obtained and future research may benefit from exploring this further, by for example, utilizing interviews with health professionals to explore these concepts in more depth.

Health professionals in the UK in the current study indicated they would see children with CRMS/CFSPID annually (29%), 6 monthly (29%) or 2–6 times per annum (16.1%). In addition, over half of health professionals in the UK in the current project (58.1%) stated children with CRMS/CFSPID were ‘sometimes’ started on CF care pathways and 45.2% stated that they would ‘sometimes’ be started on protein modulator therapies, which cost in the region of £100k per child/per year. Current recommendations suggest the frequency of clinical review for children aged ≤2 years with CRMS/CFSPID is dependent on the well-being of the infant, parental anxieties, and perceptions. Children aged between 3 and 5 years should be reviewed annually, with further reviews when the child reaches 6 years and adolescence [7]. Therefore, the actual cost of managing children with a CRMS/CFSPID designation may be a lot higher than anticipated depending on whether current recommendations or actual practice is used to calculate costs.

Health professionals in the current study indicated that picking up more children with CRMS/CFSPID could be advantageous in terms of detecting children who do become symptomatic early and therefore avoiding diagnostic delays. They also valued this information as a means of increasing knowledge about different genotypes and how best to treat them over time. Evidence for managing children with CRMS/CFSPID is still evolving; those who convert to a CF diagnosis may benefit from early intervention to prevent long term complications [7]. But it is unclear which children this applies to or how frequently they should be monitored [7,10,11]. Health professionals in the current study also felt that if additional evidence were available to inform management of children with CRMS/CFSPID, this would reduce anxiety by decreasing overmedicalization of this group of children.

## 5. Conclusions

In contrast to previous research with the UK general public [19], most participants in the current international study indicated a preference for the sensitive approach to EGS were it to be introduced into NBS for CF. This was due to the perceived importance of identifying all children with CF as early as possible following screening, to enable them to access appropriate treatment and ensure better health outcomes. In addition, identifying more children with CRMS/CFSPID was thought to be potentially beneficial by some, as it could facilitate research and an overall better understanding of how these children should best be managed. Identifying more cases of CRMS/CFSPID was viewed as potentially less physically and psychologically damaging than a missed case of CF. These views of health professionals with experience of caring for children with CRMS/CFSPID will be important to inform future decision-making around EGS.

## Figures and Tables

**Table 1 IJNS-10-00031-t001:** Professional role and whether participants had experience of caring for children with CF and their families.

	Region			Care for Children with CF			Total
Role	UK	Europe/RoW ^1^	US	No	Yes	Did Not Answer	
Centre Director	1	1	10	0	12	-	12
CF Clinical Nurse Specialist	13	0	3	2	12	2	16
CF Consultant	19	9	6	2	29	3	34
Clinical Geneticist	0	1	0	0	1	-	1
Clinical Research Coordinator	1	0	0	1	0	-	1
Dietician	1	0	0	1	0	-	1
Genetic Counsellor	2	0	5	2	5	-	7
Health and Physical Education	0	1	0	-	-	1	1
Laboratory staff	1	4	0	4	1	-	5
Newborn Screening Coordinator	1	4	1	4	2	-	6
Nurse Other	0	0	4	0	4	-	4
Nurse Practitioner	0	0	1	0	1	-	1
Pediatrician	4	4	1	0	8	1	9
Physician Assistant	0	0	2	0	2	-	2
Physiotherapist	3	0	0	0	3	-	3
Professor/Associate Professor	0	3	1	0	4	-	4
Screening Nurse	1	0	0	1	0	-	1
Total	47	27	34	17	84	7	108

^1^ Europe/RoW includes countries in Europe but not in the EU such as Norway; it also includes single participants from Israel, Turkey, New Zealand Argentina and Brazil.

**Table 2 IJNS-10-00031-t002:** Years working with children/adults/families with experience of cystic fibrosis (affected, carrier, CRMS/CFSPID designation).

	Region			
Years of Experience	UK N (%)	Europe/RoW N (%)	US N (%)	Total N (%)
5 and under	5 (11%)	5 (19%)	3 (9%)	13 (12%)
6–10	10 (21%)	2 (7%)	10 (29%)	22 (20%)
11–20	14 (30%)	4 (15%)	6 (18%)	24 (22%)
21–30	16 (34%)	12 (44%)	8 (24%)	36 (33%)
Over 30	2 (4%)	4 (15%)	7 (21%)	13 (12%)
Total	47	27	34	108

**Table 3 IJNS-10-00031-t003:** Summary of responses to questions about incorporating NGS into CF screening.

	UK	Europe/RoW	US	Total	Significance Chi Squared *
	N (%)	N (%)	N (%)	N (%)	
How important do you think it is to ensure babies with true CF are not missed?					
Neutral	1 (2.1%)	2 (7.4%)	0 (0.0%)	3 (2.8%)	-
Moderately important	10 (21.3%)	2 (7.4%)	0 (0.0%)	12 (11.1%)	df 4
Very important	36 (76.6%)	23 (85.2%)	34 (100.0%)	91 (86.1%)	**F 0.004**
How important do you think it is to reduce the number of babies reported with a CRMS/CFSPID designation by using a specific (rather than sensitive) approach if next-generation sequencing were to be implemented?					
Low/Not Important	8 (17.8%)	1 (3.8%)	7 (21.2%)	16 (15.4%)	-
Neutral	9 (20.0%)	1 (3.8%)	4 (12.1%)	14 (13.5%)	df 6
Moderately important	10 (22.2%)	7 (26.9%)	9 (27.3%)	25 (25.0%)	F 0.149
Very important	18 (40.0%)	17 (65.4%)	13 (39.4%)	48 (46.2%)	
How important would it be to use a sensitive approach?					
Low importance	3 (6.7%)	1 (4.0%)	0 (0.0%)	4 (3.9%)	-
Neutral	9 (20.0%)	5 (20.0%)	2 (6.3%)	16 (15.7%)	df 6
Moderately important	17 (37.7%)	8 (32.0%)	11 (34.4%)	36 (35.3%)	F 0.275
Very important	16 (35.6%)	11 (44.0%)	19 (59.4%)	46 (45.1%)	
How important would it be to limit reporting to use a specific approach?					
Low/Not Important	7 (15.6%)	2 (8.0%)	11 (35.5%)	20 (19.8%)	10.16
Neutral	16 (35.6%)	6 (24.0%)	8 (25.8%)	30 (29.7%)	df 6
Moderately important	10 (22.2%)	6 (24.0%)	6 (19.4%)	22 (21.8%)	0.119
Very important	12 (26.7%)	11 (44.0%)	6 (19.4%)	29 (28.7%)	
How important do you think it is to reduce or avoid the number of babies being reported as ‘probable CF carriers’ (those with one CF causing gene but may not be affected by CF)?					
Low/Not Important	6 (14.0%)	3 (12.5%)	10 (29.4%)	19 (18.8%)	24.17
Neutral	11 (25.6%)	0 (0.0%)	8 (23.5%)	19 (18.8%)	df 6
Moderately important	17 (39.5%)	5 (20.8%)	10 (29.4%)	32 (31.7%)	**<0.001 ***
Very important	9 (20.9%)	16 (66.7%)	6 (17.6%)	31 (30.7%)	

* Chi-Squared statistic, degrees of freedom and significance level. F indicates Fisher–Freeman–Halton Exact Test used. Bold text indicates a significant association.

**Table 4 IJNS-10-00031-t004:** Summary of responses to closed-ended questions about incorporating NGS into CF screening.

How Important Is It to…	UK		Europe/RoW		US		Total		Significance		
	Median (Min–Max)	IQR	Median (Min–Max)	IQR	Median (Min–Max)	IQR	Median (Min–Max)	IQR	N	K-W ^(df,2)^	*p*
…avoid missing a baby with CF	10 ^a^ (5–10)	9–10	10 (5–10)	10–10	10 ^a^ (9–10)	9–10	10 (5–10)	9–10	101	7.19	**0.028**
…avoid identifying a baby with CRMS/CFSPID	7 (2–10)	5–9	8 (1–10)	4–9	7 (1–10)	6.25–10	7 (1–10)	5–9	97	2.69	0.260
…avoid reporting carriers	6 (2–10)	5–8	8 ^b^ (1–10)	4–8	6 ^b^ (1–10)	6.5–10	7 (1–10)	5–8	91	5.76	*0.056*
…reduce repeat IRT tests	5 (1–10)	3–8	7 (1–10)	4.5–9	7 (1–10)	4–9	7 (1–10)	4–8	93	2.02	0.365

Letters indicate significant pairwise comparison (with Bonferroni correction): ^a^ = 0.022, ^b^ = 0.079. Bold text indicates significance at *p* < 0.05, italics indicate a non-significant trend *p* < 0.1.

**Table 5 IJNS-10-00031-t005:** Summary of responses concerning the impact of CRMS/CFSPID.

in Your Experience…	UK	Europe/RoW	US	Total	Significance
	N (%)	N (%)	N (%)	N (%)	
**…does a** CRMS/**CFSPID designation impact on the family in any way?**					
No	0 (0.0%)	0 (0.0%)	1 (3.2%)	1 (1.3%)	df 4 F 0.470
Sometimes	4 (12.9%)	3 (17.6%)	8 (25.8%)	15 (19.0%)	
Yes	27 (87.1%)	14 (82.4%)	22 (71.0%)	63 (79.7%)	
**…how frequently are children with a designation of CRMS/CFSPID and their families reviewed by a clinical team?**					
2 to 6 times/year	5 (16.1%)	0 (0.0%)	2 (6.5%)	7 (8.9%)	df 12 F 0.291
6 monthly	9 (29.0%)	5 (29.4%)	8 (25.8%)	22 (27.8%)	
Frequently initially then annually	1 (3.2%)	3 (17.6%)	2 (6.5%)	6 (7.6%)	
Annually	9 (29.0%)	8 (47.1%)	14 (45.2%)	31 (39.2%)	
Depends on clinical signs	4 (12.9%)	1 (5.9%)	3 (9.7%)	8 (10.1%)	
Don’t know	3 (9.7%)	0 (0.0%)	0 (0.0%)	3 (3.8%)	
Other	0 (0.0%)	0 (0.0%)	2 (6.5%)	2 (2.5%)	
**…are children with a** CRMS/**CFSPID designation started on standard CF care pathways?**					
No	13 (41.9%)	10 (58.8%)	14 (45.2%)	37 (46.8%)	df 4 *F 0.076*
Sometimes	18 (58.1%)	5 (29.4%)	17 (54.8%)	40 (50.6%)	
Yes	0 (0.0%)	2 (11.8%)	0 (0.0%)	2 (2.5%)	
**…does this include CFTR modulator therapies?**					
No	16 (51.6%)	16 (94.1%)	18 (62.1%)	50 (64.9%)	df 4 **F 0.013**
Sometimes	14 (45.2%)	1 (5.9%)	11 (37.9%)	26 (33.8%)	
Yes	1 (3.2%)	0 (0.0%)	0 (0.0%)	1 (1.3%)	

Fisher–Freeman–Halton Exact Test (degrees of freedom and significance). Bold text indicates significant association; italics indicate non-significant trend (*p* < 0.1).

## Data Availability

The data presented in this study are available on request from the corresponding author. The data are not publicly available due to ethical constraints.

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
