# Peer review of "International Perspectives of Extended Genetic Sequencing When Used as Part of Newborn Screening to Identify Cystic Fibrosis"

_2409-515X, 2024, doi:10.3390/ijns10020031_

Round 1

Reviewer 1 Report

Comments and Suggestions for Authors

The research summaries the view of a large number of health professionals on the expanded use of genetic sequencing to CF NBS.

This work is original in that it is a structured quantitative measurement of the survey responses. In addition, the international nature of the survey allows for comparisons across different countries which is a gap in the field.

It is important for the subject area because of the addition of the international, quantitative and updated survey that inform NBS programs as they make decisions on expanded genetic testing for CF in NBS.

It is not necessary for the manuscript to still be a contribution to the field, but the authors could add discussion of plans to resolve the inconsistency in responses mentioned in lines 451 - 458. For example, do the authors plan to modify the survey questions and run it again? Any attempt to contact survey participants with contradictory responses to answer the question if it was balancing desire or if it was confusion on the questions?

Some of the conclusions are supposed by the evidence. The majority of respondents did indicate they prefer sensitivity over specificity. However, I do not see quantitative support for the statement "CFSPID was thought to be potentially beneficial..."; perhaps the anecdotal evidence of responses, but this was not quantified.

For tables, perhaps shading the columns by country could make it easier to view the data.One more question - the totals in lines 110-112 total 81, but the text says 80 were excluded. 

Author Response

The research summaries the view of a large number of health professionals on the expanded use of genetic sequencing to CF NBS.

This work is original in that it is a structured quantitative measurement of the survey responses. In addition, the international nature of the survey allows for comparisons across different countries which is a gap in the field.

It is important for the subject area because of the addition of the international, quantitative and updated survey that inform NBS programs as they make decisions on expanded genetic testing for CF in NBS.

Thank you very much for taking the time to review this manuscript and thank you for your positive feedback, it is very much appreciated. Please find the detailed responses below and the corresponding revisions in track changes in the re-submitted file.

It is not necessary for the manuscript to still be a contribution to the field, but the authors could add discussion of plans to resolve the inconsistency in responses mentioned in lines 451 - 458. For example, do the authors plan to modify the survey questions and run it again? Any attempt to contact survey participants with contradictory responses to answer the question if it was balancing desire or if it was confusion on the questions?

As the survey was anonymous we cannot contact participants with contradictory responses. We currently have no plans to repeat the survey but if we did, we think it would be beneficial to include interviews with health professional to gain a more detailed understanding of how these contradictory views arise and the reasoning behind them. However, given responses to other questions, we feel this is most likely to be attributable to respondents wanting to not miss cases of CF and also reduce the number of children identified with CFSPID which is not an option via the sensitive or specific approach but clinically would clearly be desirable.  We have added the following into the discussion to address this:

“…and future research may benefit from exploring this further by for example, utilizing interviews to explore these concepts in more depth.” (line 463)

Some of the conclusions are supposed by the evidence. The majority of respondents did indicate they prefer sensitivity over specificity. However, I do not see quantitative support for the statement "CFSPID was thought to be potentially beneficial..."; perhaps the anecdotal evidence of responses, but this was not quantified.

We have clarified this by adding ‘..by some’ as this was mainly derived from the qualitative comments from the open-ended questions. (line 495)

For tables, perhaps shading the columns by country could make it easier to view the data.

We would be open to doing this but are not sure if this is permitted by the journal. We are happy to discuss options with the editors.

One more question - the totals in lines 110-112 total 81, but the text says 80 were excluded. 

This has been corrected, thank you for noticing this!

Reviewer 2 Report

Comments and Suggestions for Authors

Thank you to the authors for undertaking this interesting international questionnaire study. This reviewer opines that the content is timely and will be appreciated by IJNS readership and international membership. Several points have been provided below that this reviewer opines is necessary to address and provide sufficient clarity. Issues to be considered are below; I am happy to re-review should the authors/editors request.

Abstract: Readability is difficult from this reviewer’s perspective. 

Line 15-17 Please explain to readers what this means as many will not be familiar with terminology:

“ How this is implemented will have implications for the number of children detected with the uncertain outcome of CRMS/CFSPID (cystic fibrosis trans-16 membrane conductance regulator (CFTR)-related metabolic syndrome / CF Screen Positive Incon-17 clusive Diagnosis),…”

1.Introduction:

--Line 32-33 While the authors are experts in the field, many readers are not and will need far more background to start and especially to know what “CF transmembrane conductance regulator (CFTR)” is, and it’s significance. Please provide more introduction for readers before leaping to what was stated in this first sentence:

“ Advances in genomic sequencing technologies mean that the use of CF transmembrane conductance regulator (CFTR) extended genetic sequencing (EGS) has become feasible for cystic fibrosis (CF) newborn screening (NBS). “

To this reviewer, a facet of the concept in sentence line 45-49 may be more relevant to the less familiar reader to open up with.

--Recommend modifications for easier readability. Many sentences are overly complex in structure with multiple clauses on top of multiple terms/parens/punctuation and would benefit from breaking up into smaller sentences.

2. Materials-Methods

 Online Survey: This is lacking sufficient information herein. 

Moreover, please include a note to see supplemental for full questionnaire

--Line 80 : Please give provide details –such as how many questions? What were the questions? Please provide specific examples and/or summarize concepts in a chart --

“The second section contained questions related to the importance of using a sensitive or specific approach using a 5-point Likert response scale from very important…” 

--Line 84-6: Please give specific details of the questions and summarize concepts with  examples, perhaps in a chart --

“it included questions on the impact of a CRMS/CFSPID designation on families, and also questions relating to their experience of current CRMS/CFSPID treatment pathways.”

3. Results

Table 1:

From this reviewers perspective, Table 1 is challenging to read. 

--Suggest left justification of each “Role.”

--Consider reduction of columns (especially “care for children with CF) and instead use text in body rather than so many small cells. Then can have full table in Appendix

--Separate questions more please

Table 3:

Difficult to read; suggest delineating columns with spacing between regions

--There are too many scaled options/redundancies for the question:  “How important do you think it is to reduce or avoid the number of babies being reported as 'probable CF carriers' (those with one CF causing gene but may not be affected by CF)? “

Please spell out each scaled parameter throughout the body of the manuscript to improve readability. Please also be consistent if not included when in parens:

Line 152: “….was ‘very’, rather than ‘moderately’, important….” (i.e., “very important”; “moderately important”; etc

Line 152-155 “Although most UK participants selected ‘very’ important, a greater proportion selected moderately (very 77%: moderately 21%) compared to participants from Europe & RoW (very 85%: moderately 7%) and the US (100% very important)

From this reviewer’s perspective, it is challenging to read the results because authors’ comments blend in with the quotes. Suggest change in formatting to delineate more—such as indentation and/or italicize quotes.

Also consider bracketing the respondent info

Line 262-4: Confusing sentence structure; plus missing words: 

“Support for a specific approach was reduced to 51% of participants, when compared to the 71% of participants who had supported this option when had been framed in the  context of reducing the number of children receiving a CFSPID designation.”

Line 267-8: Confusing use of a comma here—Consider a colon

“The most frequent themes referenced in the open-ended responses to this question were, the importance of not….”

TABLE 4 Summary: 

Challenging to read. Please consider re-formatting to further delineate geographically. Consider simplifying here by placing range in parens immediately after Median rather than in it’s own column.

TABLE 5: Similar to other observations, from this reviewers perspective, Table 5 is challenging to read. 

--Suggest left justification for each issue.

--Consider reduction of columns (especially “care for children with CF) and instead use text in body rather than so many small cells. Then can have full table in Appendix

--Separate questions more please

Please check typos

Line 363: what is “pa” “2-6 times pa (9%).”

DISCUSSION

Line 415: Please clarify whether this current manuscript is part of what was sent 8 years ago?  “…was undertaken in 2016, to…”

Line 433: awkward comma: …”concerns, was…”

Line 452-58: Please break into smaller senterences

Line 486 : awkward comma: …”following screening, to enable them”

Line 490-91: Please expand Last sentence to go beyond the “result”: What do the authors think about this/conclude?  What is the relevance of this?:“Identifying more cases of CFSPID was viewed as potentially less physically and  psychologically damaging than a missed case of CF.”  

Author Response

Thank you to the authors for undertaking this interesting international questionnaire study. This reviewer opines that the content is timely and will be appreciated by IJNS readership and international membership. Several points have been provided below that this reviewer opines is necessary to address and provide sufficient clarity. Issues to be considered are below; I am happy to re-review should the authors/editors request.

Thank you very much for taking the time to review this manuscript. Please find the detailed responses below and the corresponding revisions in track changes in the re-submitted file.

Abstract: Readability is difficult from this reviewer’s perspective. 

Line 15-17 Please explain to readers what this means as many will not be familiar with terminology:

“ How this is implemented will have implications for the number of children detected with the uncertain outcome of CRMS/CFSPID (cystic fibrosis trans-16 membrane conductance regulator (CFTR)-related metabolic syndrome / CF Screen Positive Incon-17 clusive Diagnosis),…”

We have changed this so it now reads: “How this is implemented will change the number of children being given an uncertain outcome termed CRMS/CFSPID (cystic fibrosis transmembrane conductance regulator (CFTR)-related metabolic syndrome / CF Screen Positive Inconclusive Diagnosis), probable carrier results, and the number of missed CF diagnoses”

1.Introduction:

--Line 32-33 While the authors are experts in the field, many readers are not and will need far more background to start and especially to know what “CF transmembrane conductance regulator (CFTR)” is, and it’s significance. Please provide more introduction for readers before leaping to what was stated in this first sentence:

“ Advances in genomic sequencing technologies mean that the use of CF transmembrane conductance regulator (CFTR) extended genetic sequencing (EGS) has become feasible for cystic fibrosis (CF) newborn screening (NBS). “

Thank you for highlighting this. We have swapped the structure around so it now reads: “Advances in genomic sequencing technologies mean that the use of CF trans-membrane conductance regulator (CFTR) extended genetic sequencing (EGS) has be-come feasible for cystic fibrosis (CF) newborn screening (NBS). CF is caused by pathogenic variations of the CF transmembrane conductance regulator (CFTR) gene”

To this reviewer, a facet of the concept in sentence line 45-49 may be more relevant to the less familiar reader to open up with.

We believe we have addressed this by adjusting the introductory sentence as above.

--Recommend modifications for easier readability. Many sentences are overly complex in structure with multiple clauses on top of multiple terms/parens/punctuation and would benefit from breaking up into smaller sentences.

We have re-proofread the manuscript and have made modifications (including breaking up several long sentences) to improve readability throughout where necessary.

  1. Materials-Methods

 Online Survey: This is lacking sufficient information herein. 

Moreover, please include a note to see supplemental for full questionnaire

--Line 80 : Please give provide details –such as how many questions? What were the questions? Please provide specific examples and/or summarize concepts in a chart --

“The second section contained questions related to the importance of using a sensitive or specific approach using a 5-point Likert response scale from very important…” 

--Line 84-6: Please give specific details of the questions and summarize concepts with  examples, perhaps in a chart --

“it included questions on the impact of a CRMS/CFSPID designation on families, and also questions relating to their experience of current CRMS/CFSPID treatment pathways.”

We have added in the number of questions asked in each section. We have made it clear that the supplementary information contains the full survey.

  1. Results

Table 1:

From this reviewers perspective, Table 1 is challenging to read. 

--Suggest left justification of each “Role.”

--Consider reduction of columns (especially “care for children with CF) and instead use text in body rather than so many small cells. Then can have full table in Appendix

The roles were left justified at the point of submission so the authors wonder if this change was implemented by the editorial team? We are very happy to discuss the formatting of the table to improve readability with the editors, within what is allowed for the journal. 

We feel these data are easier to understand in a table format rather than using text in the body.

--Separate questions more please

Thank you for your suggestion. We have made the rows containing the questions slightly bigger, so they appear to be separated from the responses.

Table 3:

Difficult to read; suggest delineating columns with spacing between regions

Thank you for your suggestion. We would be happy to add shading as per the reviewer one's comments and will discuss this with the editors to ensure we can improve readability within what is allowable for this journal.

--There are too many scaled options/redundancies for the question:  “How important do you think it is to reduce or avoid the number of babies being reported as 'probable CF carriers' (those with one CF causing gene but may not be affected by CF)? “

Apologies, this has been removed, it was an editing error - thank you for pointing this out. 

Please spell out each scaled parameter throughout the body of the manuscript to improve readability. Please also be consistent if not included when in parens:

Line 152: “….was ‘very’, rather than ‘moderately’, important….” (i.e., “very important”; “moderately important”; etc

Line 152-155 “Although most UK participants selected ‘very’ important, a greater proportion selected moderately (very 77%: moderately 21%) compared to participants from Europe & RoW (very 85%: moderately 7%) and the US (100% very important)

Thank you we have added ‘important’ and removed the parens as suggested.

From this reviewer’s perspective, it is challenging to read the results because authors’ comments blend in with the quotes. Suggest change in formatting to delineate more—such as indentation and/or italicize quotes.

Also consider bracketing the respondent info

All quotes were italicised at the point of submission and so again, I think this may be a style change implemented by the journal. We have re-italicised the direct quotes and have added brackets to the respondents’ information as suggested (but will need to check this is acceptable with the editors).

Line 262-4: Confusing sentence structure; plus missing words: 

“Support for a specific approach was reduced to 51% of participants, when compared to the 71% of participants who had supported this option when had been framed in the  context of reducing the number of children receiving a CFSPID designation.”

Thank you, we have rephrased this sentence.

Line 267-8: Confusing use of a comma here—Consider a colon

“The most frequent themes referenced in the open-ended responses to this question were, the importance of not….”

Thank you, we have added a colon.

TABLE 4 Summary: 

Challenging to read. Please consider re-formatting to further delineate geographically. Consider simplifying here by placing range in parens immediately after Median rather than in it’s own column.

As above, we would be happy to add shading if allowable by the journal. We prefer to keep the range in a separate column, but thank you for the suggestion. 

TABLE 5: Similar to other observations, from this reviewers perspective, Table 5 is challenging to read. 

--Suggest left justification for each issue.

--Consider reduction of columns (especially “care for children with CF) and instead use text in body rather than so many small cells. Then can have full table in Appendix

--Separate questions more please

Again, the columns were originally left justified. We can discuss with the journal whether this can be returned to left justified within the scope of their style guide. There is no column ‘care for children with CF’ in Table 5 (may be a copy and paste error from comment above?)

Please check typos

Line 363: what is “pa” “2-6 times pa (9%).”

This has been changed to ‘/year’ for clarity, instead or pa (i.e., per annum)

DISCUSSION

Line 415: Please clarify whether this current manuscript is part of what was sent 8 years ago?  “…was undertaken in 2016, to…”

This was not part of the work undertaken in 2016. We have made appropriate changes to the beginning of the discussion to clarify this.

Line 433: awkward comma: …”concerns, was…”

Thank you, we have removed the comma.

Line 452-58: Please break into smaller senterences

Thank you, we have broken this into two sentences.

Line 486 : awkward comma: …”following screening, to enable them”

Thank you, but we feel this comma is needed to ensure the sentence makes sense.

Line 490-91: Please expand Last sentence to go beyond the “result”: What do the authors think about this/conclude?  What is the relevance of this?:“Identifying more cases of CFSPID was viewed as potentially less physically and  psychologically damaging than a missed case of CF.”  

We have added a sentence to conclude our thoughts.

Round 2

Reviewer 2 Report

Comments and Suggestions for Authors

Thank you for making several modifications to this manuscript to enhance understanding of this important topic and for clarifying the formatting issues. From this reviewers' perspective, it would be helpful to readers if the authors provide further clarification to Line 66: “This work” -- still unclear to this reviewer which timeframe is “this” particularly since line 65 uses "this" and references an earlier study.. Suggest either adding “present” or “current”  (i.e., ”This present work/study”) . Thanks for considering.